# Michelson Interferometric Methods for Full Optical Complex Convolution

**DOI:** 10.3390/nano14151262

**Published:** 2024-07-28

**Authors:** Haoyan Kang, Hao Wang, Jiachi Ye, Zibo Hu, Jonathan K. George, Volker J. Sorger, Maria Solyanik-Gorgone, Behrouz Movahhed Nouri

**Affiliations:** 1Optelligence LLC., 10703 Marlboro Pike, Upper Marlboro, MD 20772, USA; haoyankang@ufl.edu (H.K.); hwang40@ufl.edu (H.W.);; 2Department of Electrical and Computer Engineering, The George Washington University, 800 22nd St NW, Washington, DC 20052, USA; jiachi.ye@ufl.edu (J.Y.); huzibo@email.gwu.edu (Z.H.); jonathan.k.george@gmail.com (J.K.G.); msolyanik@gwu.edu (M.S.-G.)

**Keywords:** freespace optics, Fourier optics, 4F system, optical convolution

## Abstract

Optical real-time data processing is advancing fields like tensor algebra acceleration, cryptography, and digital holography. This technology offers advantages such as reduced complexity through optical fast Fourier transform and passive dot-product multiplication. In this study, the proposed Reconfigurable Complex Convolution Module (RCCM) is capable of independently modulating both phase and amplitude over two million pixels. This research is relevant for applications in optical computing, hardware acceleration, encryption, and machine learning, where precise signal modulation is crucial. We demonstrate simultaneous amplitude and phase modulation of an optical two-dimensional signal in a thin lens’s Fourier plane. Utilizing two spatial light modulators (SLMs) in a Michelson interferometer placed in the focal plane of two Fourier lenses, our system enables full modulation in a 4F system’s Fourier domain. This setup addresses challenges like SLMs’ non-linear inter-pixel crosstalk and variable modulation efficiency. The integration of these technologies in the RCCM contributes to the advancement of optical computing and related fields.

## 1. Introduction

Modern computing faces increasing demands for high-speed and efficient data processing, driven by the exponential growth in data generation and the complexity of computational tasks. Traditional electronic computing systems, while continuously improving, are approaching physical limitations in terms of processing speed and energy efficiency. This is particularly evident in data-intensive applications such as artificial intelligence, big data analytics, and real-time signal processing, where the large volume of information to be processed poses significant challenges. The bottleneck in these systems often lies in data movement and complex mathematical operations, especially in convolutional processes central to many machine learning algorithms. Additionally, the energy consumption and heat generation of dense, high-performance computing clusters presents both economic and environmental concerns. These challenges have spurred research into alternative computing paradigms that can potentially overcome the limitations of traditional electronic systems. Optical computing, leveraging the properties of light for information processing, emerges as a promising avenue. It offers the potential for parallel processing, reduced energy consumption, and faster computation of certain mathematical operations. However, realizing these benefits in practical systems requires overcoming several technical hurdles, including the development of efficient optical components and the integration of optical and electronic systems.

In this study, we examine the potential of optical computing to address current technological challenges in data processing methodologies. We present a comparative analysis of classical electronic convolution techniques and optical passive convolution, highlighting the specific characteristics of each approach. This comparison aims to provide insights into their respective roles and limitations within current computing paradigms. Our research focuses on the development of hardware solutions for heterogeneous data processing, specifically exploring the application of optical methods. We investigate how optical computing techniques can potentially overcome certain limitations of traditional electronic systems, particularly in the context of convolution operations [1,2,3,4,5]. These approaches, with a particular emphasis on their impact on accelerating computational tasks, represent a significant leap in computing architecture diversity [6,7,8]. Our experimental work demonstrates a convolution-based processor capable of modulating both phase and amplitude across two million pixels independently. We present the design, implementation, and performance metrics of this processor, discussing its capabilities and limitations within the context of current optical computing challenges. Our focus is on developing and analyzing an SLM-based Reconfigurable Simultaneous Optical Full Complex Convolution Module. We also investigate potential applications, including an Optical Hashing Module based on the SWIFFT algorithm [9]. This study examines the principles and applications of the free-space Fourier 4F system in optical computing. We analyze its potential advantages in data processing, particularly for convolution operations. Our investigation also addresses the current limitations in optical computing approaches. We specifically focus on the trade-offs between resolution and processing speed in optical systems and the constraints imposed by commercial spatial light modulators (SLMs) with their 60 Hz refresh rate limitation. These challenges are critical factors affecting the practical implementation and performance of optical computing systems. By identifying these limitations, we aim to provide a foundation for future research directions in overcoming these obstacles. We introduce the Reconfigurable Complex Convolution Module (RCCM), an approach for optical computing based on established theoretical principles. We compare the RCCM with existing optical convolution methodologies to assess its performance and potential advantages. Our research investigates optical real-time dynamic data processing, focusing on its application in tensor algebra operations. These operations are computationally demanding in fields such as machine learning, cryptography, and digital holography [10,11,12,13]. While electronic semiconductor hardware faces certain limitations in complex calculations [14], optical methods present alternative solutions. The optical Fourier transform and dot-product multiplication, performed passively by a single lens or metalens, potentially offer reduced computational complexity for specific applications. We acknowledge current limitations in optical computing systems, particularly the refresh rate constraints of SLMs. However, ongoing research in phase-changing materials and 2D materials indicates potential solutions to this challenge [15,16,17,18,19,20].

This paper presents a comprehensive study of the RCCM for optical computing. Our research addresses current challenges in data processing and explores the potential of optical computing to overcome these limitations. The study examines the fundamental principles underlying the RCCM, including Fourier optics and the crucial role of spatial light modulators. We present experimental results demonstrating the capabilities of our convolution-based processor, which independently modulates both phase and amplitude across two million pixels. Throughout the paper, we discuss current technological limitations and potential future research directions. By examining both theoretical foundations and practical implementations of the RCCM, this work aims to contribute to the ongoing development of optical computing technologies and their potential integration into next-generation computing systems.

## 2. Methods

In setups that require controllable amplitude and phase modulation, it is advantageous to switch between the two regimes on demand. Additionally, for several applications, such as heterogeneous computing and optical neural networks, optical image processing, and holography, achieving a full optical convolution potentially provides higher performance and more efficient information harvesting due to information pre-processing in the domain native to the data origin.

To realize a simultaneous phase and amplitude convolution in the Fourier domain with phase-only SLMs, we employ a Michelson interferometer, as shown in Figure 1a,b. An incoming table-top laser beam *Ae*^(^*^iϕ^*0) propagates through the Michelson setup, where it is split by a 50/50 beam splitter into two equivalent arms; see Figure 1a. Each component picks up a mix of pre-calculated amplitude and phase modulation from the two SLMs (Holoeye Pluto-2 VIS-096), see insert in Figure 1b, resulting in the wavefront becoming
(1)$$Ae^{i\varphi_{\left(x,y\right)}}\left[\frac{e^{i}arccosF_{\left(x,y\right)}+e^{-iarccosF(x,y)}}{2}\right]=AF_{\left(x,y\right)}e^{i\varphi_{\left(x,y\right)}}$$
where *A* denotes the constant amplitude of the incoming laser beam; and *φ* and arccos *F (x, y)* are the phase and the amplitude modulations to be imprinted by the designed setup, respectively. The phase masks are thus
(2)$$M_{\left(i,j\right)}^{\left(1,2\right)}=\varphi_{\left(i,j\right)}^{\left(1,2\right)}\pm arccosF_{\left(i,j\right)}^{\left(1,2\right)}$$

Here, indices *{i, j} represent* the pixels of a corresponding active region on each SLM, and (1, 2) correspond to the first SLM (SLM1) and second SLM (SLM2). To accomplish full optical convolution, SLM1 modulates the amplitude of an incoming table-top beam. After that, within the Fourier domain of a 4F system, transferred passively by a piece of lens, SLM2 and SLM3 further modulate the combined complex-valued phase and amplitude factors by projecting pre-calculated Fourier-transformed masks representing the 2-D distribution of the phase and amplitude combination, denoted as M. For a view of the experimental setup, refer to Figure 1b. Three computer-generated SLM patterns representing different convolution pairs are shown sequentially in this figure, and their results, acquired by a camera, are shown in the same sequence above the camera icon.

Figure 2a illustrates the simulation of the featured algorithm, while Figure 3a displays an image of the actual implementation. To ensure precision and minimize possible misalignments, all components are integrated within a robust Thorlabs 30 mm caging system, providing a stable and reliable framework for the optical configuration.

A comprehensive simulation was performed within a Python environment to replicate the propagation of coherent laser light and to examine the functionality of ideal lenses, utilizing Fresnel diffraction theory, accomplished with the Lightpipes simulator [21]. All SLM masks were pre-calculated through mathematical optical propagation, following the Fresnel diffraction formula, trigonometric functions, and Euler’s law to simulate the approximate Fourier transform in an experimental environment.

One of the critical stages of the simulation is the normalization of masks within the range of 0 to 2π. This normalization ensures that the correct phase-space representation is achieved, maintaining the integrity of the optical signals as they are processed through the system. It is important to highlight that achieving the proper convolution within the system is a complex process that necessitates the modulation of both amplitude and phase to represent the true behavior of the featured system. The successful simulation of coherent laser light propagation and the performance of ideal lenses could offer significant insights into the behavior of real-world optical systems.

Additionally, it is essential to maintain alignment accuracy at the level of the light’s wavelength, λ, along the axis of the optical path to avoid phase shifts and distortions in the interference pattern. The precision of overall misalignment, covering various alignment factors, must be within hundreds of nanometers. This encompasses the adjustment of the optical components’ tip and tilt and precise positioning in the XY plane. Moreover, precise adjustment of the focal lengths of both lenses, the angle of the beam splitter, and the management of the optical path length difference between the arms of the Michelson interferometer are crucial for achieving optimal performance. Altogether, there are 10 alignment degrees of freedom that demand precision from micro- to nanometer levels to guarantee the accurate and stable functioning of the interferometry setup.

During the alignment process, a collimated beam lights up both SLMs, followed by a Fourier lens converting the combined modulated beam into the frequency domain on the surface of the charge-coupled device (CCD) camera. This results in the appearance of a cross-shaped dot matrix on the camera. Ideally, with perfect alignment of the interferometer, all dots in the matrix should align perfectly. The presence of any misalignment is thus detected, enabling fine adjustments, along with the help of a Shack–Hartman sensor mounted in the optical path. The Shack–Hartman sensor can assist in cleaning up the wavefront prior to the convolution, reducing the alignment difficulties. The integration of piezo attenuators enhances this process by allowing for precise adjustments to the tip and tilt of the SLM mounts. All parts are caged in Thorlabs 30 mm caging systems to minimize possible misalignments; see Figure 3a for the experimental setup picture.

## 3. Results and Discussion

In Figure 3b, we report an initial experimental complex convolution result with a sequence size of 5 bits. Note that a fabricated glass laser mask temporarily substitutes the amplitude-modulating SLM1 to reduce pixel-by-pixel alignment difficulty, which requires precision within the tens of nanometers level. The plot compares the experimental and the simulated results in the central area and an exemplary mapping of both results with the smaller sequence size. It can be found from the intensity distribution that, despite an overall calculated intensity error rate of approximately 20%, all peaks are found at the estimated X-axis position, which promises the capability of performing simultaneous full complex optical modulation of the featured algorithm and schematic.

### 3.1. SLM Modulation Accuracy

Complex convolution accuracy is reduced by inter-pixel crosstalk, an inherent property of SLMs. Ideally, the optical power at 255 and 0 pixel values should be equal,, corresponding to a full continuous 2π modulation range. However, the inter-pixel crosstalk prevents this ideal behavior. This crosstalk occurs due to the non-local nature of the electric field that controls the liquid-crystal-on-silicon (LCOS) dipoles in each pixel of the modulatorDue to the drift of the electric field into the adjacent pixels, the experimental pixel-to-pixel modulation depth ends up being below the nominal 2.1π range for high-density masks. The modulation range improves as the fringes widen but stays below 100%, indicating a full 2π modulation depth has not yet been achieved.

However, the device as a whole still performs as expected on sufficiently sparse patterns (superpixel size over 16). To assess the phase modulation efficiency of the used SLMs, we introduce a specific error rate, denoted as *η*:(3)$$Ae^{i\varphi_{\left(x,y\right)}}\left[\frac{e^{i\eta }arccosF_{\left(x,y\right)}+e^{-i\eta arccosF(x,y)}}{2}\right]=Ae^{i\varphi_{\left(x,y\right)}}cos\left(\eta arccosF_{\left(x,y\right)}\right)$$
which goes back to its original form $AF_{\left(x,y\right)}e^{i\varphi_{\left(x,y\right)}}$ when *η* = 1, i.e., for the ideal modulation. The parameter η is bounded in the huyun0, 1] range. The closer it is to 1, the better the performance of the complex convolution processor that can be expected, but when *η* is less than 0.8, the measured diffraction accuracy drops below 40%.

Figure 4 presents several experimental results of the full RCCM. These show the convolution of different input sequences with the kernel 10001, using the Euler algorithm implemented on the SLMs, see Figure 4b for the schematic. Diffraction still has an essential impact on the results, with intensity gathering mainly in the center area of the optical path, leading to an overall error rate of 12% due to the claimed modulation depth discussed in the previous section. Also, when 0s are inserted in the input, there is always a latter slot gaining more intensity than being weighted as 0, see Figure 4c,d, which is related to the physical limitation of grating behaviors introduced by the said sharp edges when the patterns on SLMs are shifting from 2π to 0.

These results hold promise that the reported algorithm is an efficient way to accomplish simultaneous phase and amplitude modulation in the Fourier domain with off-the-shelf devices capable of performing complex convolution on 2 million optical channels in parallel. Such a system is highly interesting in developing optical filters and processors for numerous applications, including optical cryptography, where neglecting phase modulation can lead to impractical bit-error rates. Therefore, we set up an optical neural network to further evaluate the featured algorithm.

### 3.2. CNN Simulation Evaluation

In our comprehensive study, we delved into the intricacies of optical modulation techniques, with a special emphasis on amplitude modulation and phase modulation, primarily facilitated through the use of spatial light modulators. Our investigation led us to employ neural network simulations to scrutinize the impact of our complex modulation approach, the Reconfigurable Simultaneous Optical Full Complex Convolution Module, on the performance metrics of optical neural networks. Remarkably, we discovered that the RCCM approach considerably boosts the efficiency and accuracy of these networks. This enhancement can be primarily attributed to the sophisticated nature of complex modulation, which introduces an additional degree of freedom over traditional modulation techniques by offering a complete optical modulation schema. This approach allows for more nuanced and precise control over the light’s properties, which is paramount in optical computing.

In our investigation, we conducted neural network simulations to assess the impact of our innovative complex modulation approach on the efficacy of optical neural networks. The findings were unequivocal: RCCM significantly amplifies the capabilities of these networks. This enhancement can be traced back to the comprehensive nature of complex modulation, which integrates both amplitude and phase modulation in a single framework. This integration provides a crucial additional degree of freedom, enabling more sophisticated manipulation of the optical signals than is possible with traditional modulation methods that treat amplitude and phase separately. The architecture of our simulated neural network was designed to include one optical layer followed by two fully connected layers, facilitating a comparison across three modulation strategies: complex modulation, amplitude modulation, and phase modulation. Our methodology involved the use of the CIFAR-10 dataset, a standard benchmark in machine learning for evaluating object recognition algorithms, to train and test the models. As the results shown in Figure 5, thecomplex modulation achieved an accuracy rate of 51.98%, surpassing amplitude modulation and phase modulation, which posted accuracy rates of 50.83% and 47.87%, respectively.

This differential in performance underscores the superiority of the RCCM approach, particularly its ability to exploit the multidimensional nature of light for computational purposes fully. By leveraging the inherent properties of light in a more holistic manner, complex modulation facilitates a higher-dimensional space for information encoding and processing, leading to more efficient and accurate neural network operations.

Moreover, these results highlight the potential of RCCM to serve as a cornerstone for the next generation of optical neural networks, offering a promising avenue for achieving high-speed, high-accuracy computations in an energy-efficient manner. The findings not only validate the theoretical advantages of complex modulation but also underscore its practical utility in enhancing the performance of optical neural networks, paving the way for its adoption in cutting-edge computational technologies [13,22,23,24].

In the development of optical neural networks, manipulating the optical field typically involves altering either the phase or the amplitude independently. However, the complete optical field encompasses both phase and amplitude, presenting an additional degree of freedom. This holistic approach allows for more intricate control and interaction within the optical system, potentially leading to significant improvements in network performance. By leveraging both parameters simultaneously, researchers can achieve higher precision and flexibility in the modulation of light, which is crucial for the complex operations required in advanced optical neural networks. The inclusion of more adjustable parameters enhances the network’s ability to learn and adapt, as it can represent and process a wider range of signal variations and complexities. Furthermore, the modulation of the complete optical field aligns more closely with the principles of electrical neural networks, particularly in the convolution process within the Fourier domain. In electrical networks, complex modulation is essential for accurately performing convolutions, a fundamental operation in many neural network architectures. By mimicking this approach in optical networks through the use of complex field modulation, the potential for achieving similar levels of performance and functionality is significantly increased. This integrated manipulation of both phase and amplitude could thus be a pivotal factor in enhancing the capabilities and efficiency of future optical computing technologies.

## 4. Applications

As per the simulation and experimental results, the proposed RCCM is a powerful developing tool for accelerating computationally algorithms, realizing the Fourier transformation in optics passively by using a single lens. We can foresee many applications based on this employment. One of the possible implementations of the featured algorithm is in Montgomery modular multiplication [25,26]. Montgomery modular multiplication is a computational method that efficiently performs modular multiplication operations of extremely large integers [27,28]. Here, *a*, *b*, and *c* denote large integers, *R* is chosen to be a power of two greater than *m*, *a*′ = *aR* mod *m* and *b*′ = *bR* mod *m*. *M* = −m^−1^ mod *R* and *c*′ denote the corresponding calculated modular production in the Montgomery domain. With the employment of RCCM, the calculation of Equation (4) can be accelerated significantly, only limited by the refresh rate of the SLMs (and resolution) [29].
(4)$$c^{\prime }=\frac{a^{\prime }b^{\prime }+\left(a^{\prime }b^{\prime }M\;modR\right)m}{R}$$

Moreover, we introduce a pioneering approach to data processing and security, focusing on optical hashing and compression techniques inspired by the SWIFFT hashing algorithm, a notable contender in the NIST SHA-3 competition [9]. Traditional computing hardware, particularly in electronic domains, faces significant challenges such as delays and high energy consumption, issues that are exacerbated in heterogeneous systems like electronic–photonic accelerators due to inefficient domain crossings [30,31,32,33].

Our approach leverages the inherent advantages of free-space optical processing, such as parallel computation, rapid tensor multiplication, and efficient Fourier transformations. These characteristics make optical systems well suited for complex data processing tasks. The RCCM utilizes these advantages to potentially enhance processing speeds significantly compared to traditional optical machine learning accelerators. This improvement is achieved by replacing conventional high-resolution cameras with faster, signal-triggered CMOS detector arrays. The RCCM’s capability for simultaneous phase and amplitude modulation enables several potential applications. In image recognition, it could process multiple images concurrently by encoding them onto different regions of the wavefront, potentially increasing throughput in large-scale recognition tasks. For data security, the RCCM could implement advanced optical hashing algorithms, utilizing both amplitude and phase information to create more robust encryption schemes, as shown in Figure 6. This approach could enhance data compression efficiency and security, offering new possibilities in optical computing and secure communications. In the realm of large-scale convolution computing, the RCCM’s ability to perform complex convolutions optically could significantly reduce the computational load on electronic systems. This feature is particularly valuable for applications in artificial intelligence and signal processing, where convolution operations are prevalent and computationally intensive. The integration of these functionalities—multi-image recognition, secure hashing, and efficient convolution—represents a convergence of machine learning and optical data processing. This integration could lead to more efficient and secure data handling in various fields, from autonomous systems to financial technology. For future directions, we are exploring two promising directions for further development. First, we are investigating the potential of self-designed indium tin oxide (ITO)-based SLMs, which could significantly increase the system’s operating speed and efficiency. Second, we are considering the transition of RCCM functionality to photonic integrated circuit (PIC) platforms. Recent advancements in PIC technology offer various options for implementing our system on a more compact and scalable platform. These developments could extend the RCCM’s capabilities to other computation-intensive algorithms, such as those used in deep learning neural networks. We aim to push the boundaries of optical and photonic computing, addressing the growing demands of data processing in the digital age with innovative and secure solutions. This approach not only promises to enhance computational capabilities but also offers potential benefits in energy efficiency and processing speed for a wide range of applications.

## 5. Conclusions 

In this study, we addressed the engineering challenge of implementing reconfigurable complex wavefront modulation for optical convolution through the development of the RCCM. Our approach focused on overcoming the limitations of current optical computing systems, particularly in terms of processing speed and versatility. The RCCM demonstrates the capability to independently modulate both phase and amplitude over two million pixels in real time. Our experimental results show that the RCCM can effectively perform complex convolutions through the interference of spatially modulated wavefronts in both phase and magnitude. In neural network simulations using the CIFAR-10 dataset, we achieved a top accuracy of 51.98%, outperforming traditional amplitude-only (50.83%) and phase-only (47.87%) modulation techniques. These results indicate the potential of our approach in improving the efficiency and accuracy of optical neural networks.

Critical analysis of our results reveals both strengths and limitations of the RCCM. The primary advantage lies in its ability to perform complex convolutions optically, potentially reducing the computational load on electronic systems. However, the current implementation faces a significant limitation in terms of operational speed, primarily due to the 60 Hz refresh rate of the commercial SLMs used in our setup. The RCCM’s capability for simultaneous phase and amplitude modulation opens up possibilities for multiple image recognition tasks. By leveraging the parallel processing nature of optical systems, the RCCM could potentially handle multiple input images simultaneously; each modulated onto different spatial regions of the wavefront. This parallelism could significantly enhance throughput in large-scale image recognition applications. Furthermore, the RCCM shows promise in the realm of encrypted data transmission, particularly in implementing optical hashing algorithms based on SWIFFT. The ability to perform complex wavefront modulation allows for the implementation of more sophisticated optical encryption schemes. By encoding data in both amplitude and phase, we can potentially create more secure optical hashing methods, enhancing data security in optical communication systems. Future work will focus on overcoming the current speed limitations of our system. We aim to explore faster modulation technologies, including novel materials for high-speed spatial light modulation. Additionally, we will investigate the integration of our RCCM technology with PICs to enhance compactness and scalability.

In conclusion, the proposed RCCM shows great promise in advancing optical computing, particularly for complex data processing, image analysis, and secure communications. Our system’s capacity to adjust both light intensity and phase across two million points simultaneously marks a significant advance in optical data handling. Ongoing research into self-designed ITO-based SLM could push the RCCM’s operating speed into the GHz range. Such improvements may dramatically enhance processing speed and efficiency, enabling real-time data analysis in fields like machine learning and cryptography. This work establishes a strong basis for future developments in adaptable optical computing systems, paving the way for more efficient data processing solutions as computational needs evolve.

## Figures and Tables

**Figure 1 nanomaterials-14-01262-f001:**
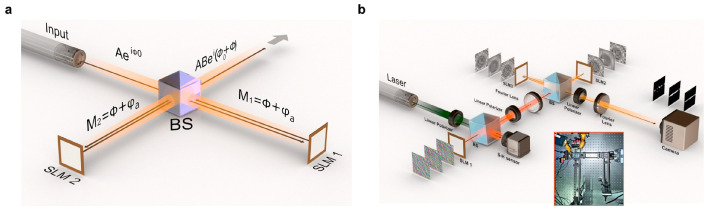
Schematic and working principle of a complex optical 4F convolution module. (**a**) A detailed schematic representation illustrating the complex convolution process based on Euler’s formula, emphasizing the mathematical underpinnings and optical path integration. (**b**) An expanded diagram of the experimental setup for executing full optical convolution. SLM1 modulated the illuminating beam’s amplitude, and SLM2 and 3, as a whole, sitting in the Fourier domain, modulated the beam with phase and amplitude controlled simultaneously; see Equation (1).

**Figure 2 nanomaterials-14-01262-f002:**
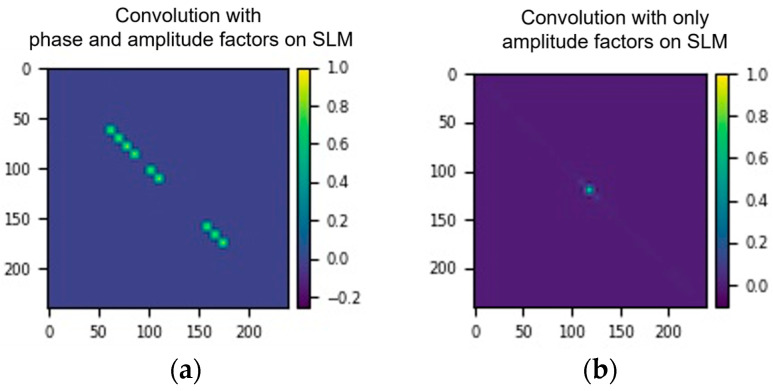
Simulated results demonstrating the impact of phase factors in the modulation device. (**a**) Simulated convolution incorporating both phase and amplitude factors, showcasing the comprehensive modulation capabilities and their combined effects on the resulting optical pattern. (**b**) Simulated convolution featuring only amplitude factors on the spatial light modulator (SLM), illustrating the distinct optical outcomes when phase factors are absent, thereby emphasizing the unique role of amplitude modulation in the convolution process. This comparison shows the necessity of simultaneously controlling both the phase and amplitude for the featured optical convolution module.

**Figure 3 nanomaterials-14-01262-f003:**
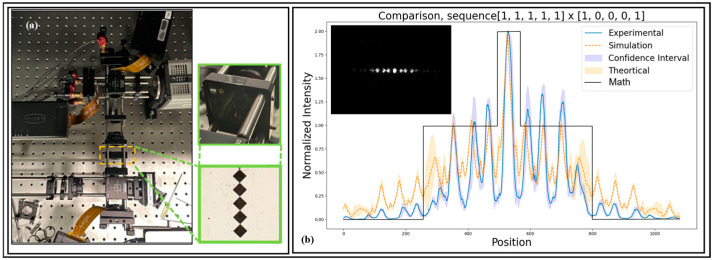
Experimental setup and basic performance metrics of a complex optical 4F convolution module. (**a**) A detailed photograph of the experimental setup, highlighting the zoomed-in area showcasing the fabricated amplitude mask and its precise installation within the system. This component is critical for modulating the light patterns as part of the convolution process. (**b**) A plot of the normalized intensity measurements acquired at the camera, juxtaposed with corresponding simulation data that include a confidence interval. This graph illustrates the convolution of the incoming sequence huyun11111] with the mask huyun10001], with the anticipated nine-digit result huyun111121111]. An inset image displays the actual output captured by the camera, providing a visual confirmation of the simulation accuracy and the experimental effectiveness. The theoretical shade represents the waveform of the classical convolution results of the sequence, and the math line represents the expected results of huyun111121111]. This figure shows evidence of the feasibility and accuracy of the RCCM in an experimental manner of a specific sequence.

**Figure 4 nanomaterials-14-01262-f004:**
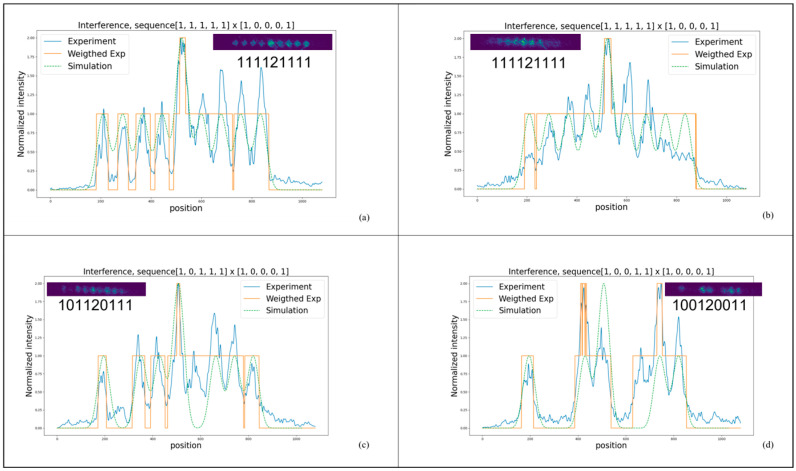
Optical convolution dynamics in a reconfigurable complex convolution module (RCCM) (**a**) Full experimental results of the RCCM: Convolution of the sequence 11111 with the kernel 10001 displayed on the spatial light modulator (SLM). This panel illustrates the primary convolution process, showcasing the direct optical output. (**b**) Adjustments with polarization-angle tuning: This modification highlights the effects of polarization-angle adjustments on the convolution output, demonstrating the influence of optical properties on the result. (**c**) Convolution result of sequence 10111 with kernel 10001: A comparison showing how minor variations in the input sequence affect the convolution outcome, providing insights into the system’s sensitivity and response. (**d**) Convolution of sequence 10011 with kernel 10001: This panel displays another variant, emphasizing how changes in the initial sequence modify the resulting convolution patterns, illustrating the dynamic capabilities of the convolution system.

**Figure 5 nanomaterials-14-01262-f005:**
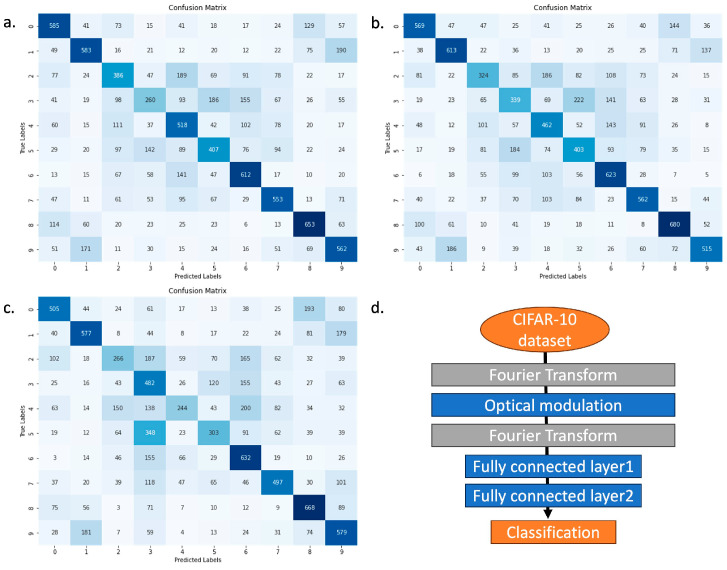
Detailed comparison of modulation techniques in CIFAR-10 neural network performance. (**a**) Complex modulation (RCCM) performance: Achieved a top accuracy of 51.98%, highlighting the effectiveness of combining amplitude and phase modulations in neural network applications. Color intensity indicates prediction frequency, with darker blue representing higher frequency and lighter shades lower frequency. (**b**) Amplitude modulation performance: Reached an accuracy of 50.83%, demonstrating its capability, though slightly less effective than complex modulation. (**c**) Phase modulation performance: Yielded the lowest accuracy of 47.87%, indicating its impact and limitations when used independently in neural networks. (**d**) Neural network architecture: Depicts the detailed architecture of the simulated neural network, providing insights into the structural elements that contribute to the performance differences observed in the modulation techniques.

**Figure 6 nanomaterials-14-01262-f006:**
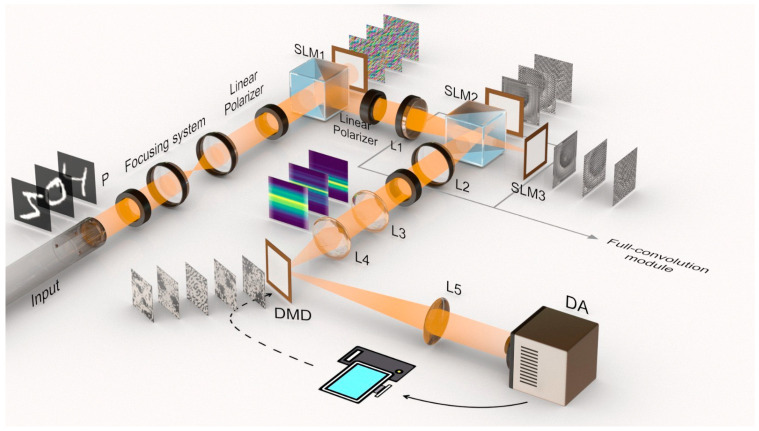
Concept of optical hashing algorithm and the scheme of the proposed full optical setup using RCCM for optical Fourier full convolution accelerator. An example of a 1-D compressed sequence read of the simulated camera reading that corresponds to MNIST input ”5”, ”0”, and ”4”; The scheme of the proposed optical setup, where the source can be a single-mode fiber array collimated into free space or coupled in a photonic chip; diffusion can be achieved by an optical amplitude-only modulation by an SLM; convolution must be performed with complex amplitude-and-phase convolution in the Fourier domain; the optional final step of image classification with a heterogeneous convolutional 4F classifier can be alternatively replaced by a CMOS diode array (DA). All inserted figures along the axis show what pattern is presenting on the SLMs or the current status of the modulated signal carried by the beam.

## Data Availability

The raw data supporting the conclusions of this article will be made available by the authors on request.

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
