# Peer review of "Michelson Interferometric Methods for Full Optical Complex Convolution"

_nanomaterials, 2024, doi:10.3390/nano14151262_

Round 1

Reviewer 1 Report

Comments and Suggestions for Authors

I have carefully read the manuscript. The idea of the work is relatively original. However, some things need to be corrected to make the manuscript clearer:

1) The use of an SLM to generate convolution is not new, but has already been used. The novelty of your work lies in having used a Michelson interferometer with two SLMs. This is not immediately evident from the text and must emerge clearly. I invite the authors to add a paragraph that clearly reports the progress of the research compared to the state of the art.

2) figure 1 is very interesting but absolutely not explanatory: the authors must modify figure 1 to explain exactly what happens along the arms of the interferometer (what are the mappings mentioned in the right figure? how does the interferometer work?)

3) I find the expression "detailed schematic" (in the caption of fig.1) misleading and contradictory: a schematic representation is not detailed..... figure 1 requires a detailed and explanatory representation, not a schematic one!

3) What is the complete scheme of the network?

4) Is the network capable of carrying out recognition or decision-making processes? How did you rate it? How is the information saved?

5) The number of pixels you can evaluate depends on the characteristics of the SLM, and is therefore limited by its technology. Do you think there is a way to increase this dimensionality by separating it from that of SLMs?

6) You talk about a learning network, but in the text it is not clear how learning occurs. Typically a neural network must be trained on multiple inputs and then verified.... none of this is present.....

What does the acronym SWIFTT- on page 2 mean?

Reviewer 2 Report

Comments and Suggestions for Authors

The authors have demonstrated a novel way to make an SLM with both amplitude and phase modulation. 

They should compare it to the alternative approach detailed in “Arbitrary manipulation of spatial amplitude and phase using phase-only spatial light modulators”, Long Zhu and Jian Wang, Scientific Reports vol 4 7441 (2014)

The usefulness of the system is doubtful. Limitations of the accuracy of the SLM lead the authors to demonstrate only the convolution of short one-dimensional binary sequences represented in sparse patterns by a fixed input. This Is a long way from the megapixel parallel operations proposed by the authors on page 7. Given that the process still involves the electronics of the spatial light modulators and cameras, it is hard to see how optical computations can outperform fully electronic methods, given the performance of constantly improving electronic computers with GPUs and their storage systems.

The claims about the advantages of optical computing (high speed, energy efficiency, accuracy) are old, and have not become real, except maybe in very limited niches. Optics is inherently analog so is constantly struggling for accuracy in a way that digital electronics does not have to. Interferometric devices are subject to stringent alignment tolerances and have significant packaging and stability challenges, as admitted by the authors.

The authors CNN simulation does not support the usefulness of their device. In no case do they achieve the high accuracy levels (~90%) achieved by ResNets on CIFAR-10. The authors write “The findings were unequivocal: RCCM significantly amplifies the capabilities”. However, the demonstrated differences in performance by the authors’ algorithm for phase only, amplitude only, and full amplitude and phase are only 4%.

The other applications: Montgomery Multiplier and optical hashing are barely introduced, and no experimental results are given to support the authors claim that they will be able to surpass the capabilities of current electronic hardware. 

Many of the references are only tangentially related to the research presented here: most are about optical computing in general.

In summary, the authors do present a novel device, but need to place it in better context and refrain from making unsupported overenthusiastic claims. Such claims unnecessarily reduce the paper’s impact.

Reviewer 3 Report

Comments and Suggestions for Authors

In the paper 'Michelson interferometric methods for full optical complex convolution' the authors present their recent results about Reconfigurable Complex Convolution Module. The paper is well written and well organized. The results are clearly presented. I have only few minor/marginal questions to the authors:

1) I think in the text it should be expanded the way in which is maintained the alignment accuracy which is fundamental for this set-up

2) in figure 3 it should be better enphasize the difference between 'math' and 'theoretical' lines.

Reviewer 4 Report

Comments and Suggestions for Authors

This is a single-blind review of manuscript nanomaterials-3018079 titled "Michelson interferometric methods for full optical complex convolution" by Haoyan Kang et al.

The work presented in the manuscript consists of the presentation of an optical setup and some measured and simulated results that indicate some potential for optical signal-processing. The setup uses a pair of commercial phase-only SLMs and some masks (?) in a Michelson interferometer configuration, and also a laser, a camera and some optomechanics (lenses, positioning control, etc.). The whole concept is based on Fraunhofer diffraction, i.e., the camera capturing a field profile that is the Fourier-transform of the aperture (the masks). 

The paper is not suitable for publication as it is mostly a "wish list" and a jumble of incoherent things. All presented results are seemingly based on well known devices and theories, with no solid indication of a novel/promising/impactful concept or technology; also, all its findings/claims appear trivial, e.g., if SLMs and/or cameras are faster then the all-optical processing is faster, or that complex-modulation (masks and phase-SLMs) is better than amplitude-only (masks) and phase-only (SLM) modulation.

A. The introduction (which is overly verbose) promises that the paper will offer a comparative analysis and uncover unique advantages etc., which never happens.

B. The schematics (e.g. Fig. 1) are not descriptive of what the setup actually is or does: What is the input profile? Where are the masks? What are the three holographic "noisy" patterns behind each SLM in panel 1b? Also, complicated setups, like Fig 1b or Fig. 6, cannot be followed or understood, drawing from the previously (not) presented ones. In addition to that, the results (e.g. Fig. 2,3,4,5) are presented with little to no context so that the reader can interpret them; the readers are not told what is attempted nor what is measured (e.g., in Fig. 2), so they cannot understand if "it worked" and how well... There are no metrics formally defined, so as to quantitatively evaluate the results.

C. Having most of the results presented in a "Discussion" section is weird. Also, that "Discussion" section not being the penultimate one (before conclusions/summary section), is weird. Shouldn't the "Applications" come before the discussion? I recommend structure: Intro > Methods > Results > Applications/Discussion > Conclusion.

D. Neural networks and modular multiplications are mentioned at some point but, again, no theoretical context or coherent results are given. For instance, Eq. (4) -- What is R and how does c' compare to c?

E. Most of the paper is written like a proposal, i.e., in a non-technical style, while some sections have an evident language problems (grammar, syntax, vocabulary, typos).

Comments on the Quality of English Language

Most of the paper is written like a proposal, i.e., in a non-technical style, while some sections have an evident language problems (grammar, syntax, vocabulary, typos).

Round 2

Reviewer 2 Report

Comments and Suggestions for Authors

Thank you for explaining your point of view: I am happy with the paper now.

Author Response

Thank you for your valuable suggestions, which are essential to us.

Reviewer 4 Report

Comments and Suggestions for Authors

This is a single-blind review of once-revised manuscript nanomaterials-3018079 titled "Michelson interferometric methods for full optical complex convolution" by Haoyan Kang et al.

The Authors have addressed part of the problems pointed out in my first review, improving the comprehensibility of the manuscript. While the overall picture is now clearer (i.e., the proposed device was shown to perform convolutions via interference of spatially modulated wavefronts, in both phase and magnitude), I still fail to understand some of the claims made, e.g.: How quantum mechanics are involved (as mentioned in the abstract and conclusions), or how Eq. (4) can be computed by this setup, or how the setup was used in the CNN benchmarking (I assume Fig. 5 data is involved, but it is neither explained nor readable).

Finally, concerning the argument about "scientific vs engineering" papers brought upon by the Authors: All papers should be technically sound, understandable, reproducible, and should follow a specific pattern (vision, targeted breakthrough, challenge, methods, results, conclusions). Now, the problem with this paper is that it spends most of its energy (typed text and vocabular eloquence) in praising the work performed in and overselling/pitching the (well known) potential benefits of this technology. This Reviewer would much rather the Authors focused on one *engineering* problem (e.g., how to efficiently implement reconfigurable complex wavefront modulation and how to put it to the use of optical convolution), then rigorously presenting the methodology to be used (and background theory), thoroughly outlining the experiments performed (and validating simulations), and critically interpreting the results. I would urge the Authors to read some of the papers by the Engheta, Alu et al. groups on analog optical signal processing, and also to compare their conclusions against the classical standards, e.g., in Goodman's Fourier optics book; what progress from that is actually presented here?

Comments on the Quality of English Language

While the writing style obviously didn't change, there's still many typos to be fixed (specifically correct capitals where lowercase should be used).

Round 3

Reviewer 4 Report

Comments and Suggestions for Authors

I would like to thank the Authors for the clarifications in the Response Letter. I would recommend incorporating some of that text in the paper as well.

Also, the abstract, intro and conclusions now read more smoothly.

No more comments.